# Comprehensive Literature Review of Hyperpolarized Carbon-13 MRI: The Road to Clinical Application

**DOI:** 10.3390/metabo11040219

**Published:** 2021-04-03

**Authors:** Michael Vaeggemose, Rolf F. Schulte, Christoffer Laustsen

**Affiliations:** 1GE Healthcare, 2605 Brondby, Denmark; Michael.vaeggemose@ge.com; 2MR Research Centre, Department of Clinical Medicine, Aarhus University, 8000 Aarhus, Denmark; 3GE Healthcare, 80807 Munich, Germany; schulte@ge.com

**Keywords:** hyperpolarized carbon-13 MRI, review, clinical application

## Abstract

This review provides a comprehensive assessment of the development of hyperpolarized (HP) carbon-13 metabolic MRI from the early days to the present with a focus on clinical applications. The status and upcoming challenges of translating HP carbon-13 into clinical application are reviewed, along with the complexity, technical advancements, and future directions. The road to clinical application is discussed regarding clinical needs and technological advancements, highlighting the most recent successes of metabolic imaging with hyperpolarized carbon-13 MRI. Given the current state of hyperpolarized carbon-13 MRI, the conclusion of this review is that the workflow for hyperpolarized carbon-13 MRI is the limiting factor.

## 1. Introduction

MR spectroscopy (MRS) and spectroscopic imaging (MRSI) obtain metabolic information noninvasively from nuclei spins. For in vivo applications, common MR-active nuclei are protons (^1^H), phosphorus (^31^P), carbon (^13^C), sodium (^23^Na), and xenon (^129^Xe). The most common are protons due to their high gyromagnetic ratio and natural abundance in the human body. Since most metabolic processes involve carbon, ^13^C spectroscopy is a valuable method to measure in vivo metabolism noninvasively [1,2,3]. ^13^C spectra are characterized by a large spectral range (162–185 ppm), narrow line widths, and low sensitivity due to the low gyromagnetic ratio (a quarter as compared to protons) and natural abundance of 1.1% in vivo. However, the sensitivity can be increased with the use of ^13^C-enriched agents and by hyperpolarization.

Hyperpolarized (HP) ^13^C MRI is a method that magnetizes ^13^C probes to dramatically increase signal as compared to conventional MRI [3]. Metabolic and functional HP ^13^C MRI is a promising diagnostic tool for detecting disorders linked to altered metabolism such as cancer, diabetes, and heart diseases [4], increasing sensitivity sufficiently to map metabolic pathways in vivo without the use of ionizing radiation, as in positron emission tomography (PET) imaging. Metabolic imaging using HP ^13^C compounds has been translated successfully into single-organ examinations in healthy controls and various patient populations [5,6,7,8,9,10].

This review aims to address the current status of HP ^13^C MRI, based on the literature from the last two decades, and provide a comparison across multiple anatomical applications, highlighting the future directions needed to elevate the method for more widespread adoption in clinical practice.

## 2. Literature Search and Review Strategy

The papers in this review include HP ^13^C MR studies from the initial publications in 2003 up to December 2020. Furthermore, the papers are PubMed (www.pubmed.gov, access on 31 December 2020) indexed, accessible online, and written in English. Selected authors in the field were searched, and forward and backward citations of retrieved studies were checked to investigate further relevant studies. Highlighted areas had to include at least 25 papers to be considered as a major topic in this review. Topics with fewer papers are included in the Technical Advances or Other sections.

An illustration of the selection process can be observed in the PRISMA 2009 Flow Diagram (Figure 1). Records with no or only limited HP ^13^C contents were excluded (Table A1). This resulted in the exclusion of 280 papers in the initial screening process. To evaluate the eligibility of the studies, current review papers were rated higher for inclusion. From a total of 1094 identified studies, 145 (13%) met the inclusion criteria for this review paper; furthermore, 12 studies not covering hyperpolarized carbon-13 MR were included.

This review does not have a predefined protocol and was conducted by invitation from editors in the *Metabolites* Special Issue entitled: “Applications of Magnetic Resonance (MR)-Based Metabolic Imaging in Medicine”.

## 3. Hyperpolarized Carbon-13 MRI

^13^C-enriched compounds for metabolic imaging studies are most commonly hyperpolarized via dynamic nuclear polarization with subsequent dissolution (d-DNP) (Figure 2) [11]. A sample of the probe (typically (1-^13^C) pyruvate) and a radical with an unpaired electron is placed in a 0.8 K cold environment at 5 T within the hyperpolarizer (e.g., SPINlab (GE Research Circle Technology Inc., GE Healthcare, Chicago, IL, USA)). These unpaired electron spins are polarized to nearly 100% at this temperature and magnetic field strength. The sample is then irradiated at the electron spin resonance frequency (e.g., 140 GHz at B0 = 5 T) to transfer the high polarization from the electron spins to the less polarized ^13^C-enriched molecules of the sample, which typically takes 30–180 min [8,12,13]. Following this procedure, the sample is then rapidly dissolved in a hot water solution to obtain an injectable solution matching the body pH, temperature, and osmolarity before injection. After a final quality assessment, the hyperpolarized solution can be injected into the subject, preferably via an injector. Following the injection of the probe, optimized fast MR sequences for the targeted organs image the uptake and subsequent metabolic conversion of the hyperpolarized ^13^C probe. The hyperpolarized probes are diluted heavily in the body and relax due to the spin-lattice (T1) decay. Radio-frequency (RF) excitation and metabolic conversion lead to further signal depletion. A hyperpolarized experiment is typically completely relaxed within minutes (1–2 min). Due to low concentrations, multiple hyperpolarized scans could be performed in the same scan session. Nevertheless, the polarization process requires up to 180 min, and the polarizer, e.g., the SPINlab, has a maximum capacity of four units. In combination with the cost of the hyperpolarized probes and ethical considerations, experiments are often limited to injecting a single dose.

The list of HP ^13^C probes currently includes more than 24 different metabolites. A description of their chemical structures is given by Keshari et al. [14]. A description of the T1, chemical shift, applications, metabolic, and physiological processes of the most common probes has recently been published in a paper by Wang et al. and is considered outside the scope of this paper [12].

## 4. Applications of Hyperpolarized Carbon-13 MRI

### 4.1. Hyperpolarized [1-13C]pyruvate: The Most Used Biomarker

Following injection into the bloodstream, the hyperpolarized [1-13C]pyruvate is transported to the tissue of interest, whereafter the [1-13C]pyruvate is transported into the cells, mediated by the monocarboxyl transporters (MCTs). In the cytosol, the pyruvate is then either enzymatically converted into lactate via lactate dehydrogenase (LDH) or alanine via alanine aminotransferase (ALT) or further transported into the mitochondria where it undergoes another enzymatic exchange reaction into CO_2_ via pyruvate dehydrogenase (PDH). The CO_2_ is then rapidly converted into bicarbonate via carbonic anhydrase (Figure 2). The glycolytic activity in the cytosol is estimated by (1) the [1-^13^C]pyruvate conversion to [1-^13^C]lactate via the enzyme LDH and (2) mitochondrial TCA cycle activity, determined by the irreversible conversion of [1-^13^C]pyruvate to bicarbonate (HCO_3_^-^) by the enzyme PDH [15].

### 4.2. Oncology

The existing publications with clinical trials of HP ^13^C MRI are dominated by oncologic applications because this technology is particularly well suited for studying cancer. Elevated glycolysis and thus lactate production even under sufficient oxygen availability, denoted by the Warburg effect [16], is a fundamental property of many cancers [17]. This phenomenon is indicated by an upregulation of the pyruvate to lactate conversion [18]. Therefore, imaging of the metabolic conversion of [1-^13^C]pyruvate into [1-^13^C]lactate holds particular promise for cancer diagnosis as well as monitoring of response to treatment [19]. The first clinical ^13^C study targeted prostate cancer in 2013 [7]. As of today, [1-^13^C]pyruvate has been applied clinically in several different cancer types ranging from prostate [7,7,20,21,22,23,24,25,26], pancreas [27], breast [28,29,30], brain [9,31,32], to kidney [33]. However, several studies are on the way as indicated by assessment of studies on clinical.trails.gov (Table A2) and EudraCT (Table A3). Future clinical studies might include probes such as urea (perfusion) and glutamine [1].

### 4.3. Brain

The metabolic imaging of the brain has been applied in multiple sclerosis [34], stroke [35], traumatic brain injury [36,37,38], and brain tumors (mentioned in the Oncology section). Brain metabolism must consider the blood–brain barrier (BBB), which limits the uptake into brain tissue of hyperpolarized probes and thus ultimately the obtainable signal in the brain. This has been an obstacle for pre-clinical studies in which the animals are anesthetized [39,40,41]. Conventionally, anesthetics are not needed in brain studies in humans; nevertheless, in some cases of intensive care patients, children (2–10 years old), or claustrophobic patients it may be preferred to apply sedation prior to the examinations, though this method can complicate metabolic response. Brain studies using multichannel receiver head coils can increase cortical signal at the expense of inhomogeneous receiver profiles and less signal-to-noise ratio (SNR) in the center of the brain [42,43,44,45]. Clinical trials of the brain (non-cancer) are reported in three studies on healthy brain (n = 4 [46]; n = 4 [13]; n = 14 [8]).

### 4.4. Cardiovascular Disease 

The MRI of cardiovascular diseases commonly evaluates the restriction of blood flow and ischemic areas of the heart [47]. It is, however, well established that the metabolic balance between the fat and sugar utilization of the heart is important in determining the underlying pathophysiology and best treatment for the individual patient [48]. HP ^13^C MRI has been shown to measure the metabolism and perfusion of the heart [49], which can be advantageous for evaluation of myocardial complications associated with diabetes, ischemic heart disease, cardiac hypertrophy, and heart failure [50,51]. The ability to image in rapid succession ensures that HP ^13^C MRI can be incorporated in stress test imaging sessions without adding significant time to these protocols. Cardiac imaging protocols need to consider cardiac cycle timing, motion correction, distortion correction, etc. [52,53]. To date, only a few clinical studies have been performed on the heart covering initially normal hearts (n = 4) [5] and later hearts of patients with type 2 diabetes (T2MD = 5, HC = 5) [54]. Evaluation of the pre-clinical literature supports a growing intention for the transition of HP ^13^C MRI towards cardiac applications [55].

### 4.5. Kidney Disease

There is a lack of good biomarkers for early diagnosis, patient stratification, and treatment monitoring for kidney diseases. MRI is increasingly being used to characterize important pathophysiological processes such as perfusion, fibrosis, and oxygenation [56,57].

Hypoxia is a hallmark of kidney disease, and thus metabolic imaging techniques able to depict either pO_2_ directly or the indirect effect of hypoxia are warranted. Hyperpolarized [1-^13^C]pyruvate studies have been demonstrated to allow differentiation of various renal pathophysiological conditions in pre-clinical models of diabetes, acute kidney disease (AKI), and chronic kidney disease (CKD) [58,59,60,61]. The use of gadolinium-based contrast agents is contraindicated in patients with renal insufficiency [62]. Thus, alternative non gadolinium-based biomarkers to noninvasively determine hemodynamic properties, perfusion, and glomerular filtrations constitute a valuable tool for a patient group in which repeated exposure to ionizing radiation is a concern [63,64,65]. Currently, only one non-cancer human kidney study is underway.

From the pre-clinical indications, it is very likely that metabolic and functional imaging with HP ^13^C MRI will be a future diagnostic tool in kidney disease. The results are promising in the application of [1-^13^C]pyruvate and have shown potential with other carbon-based probes [1,4-^13^C]fumarate, ^13^C-urea [66] and [1-^13^C]lactate [67]. Alternative biomarkers with improved relaxation properties also hold great promise in perfusion assessment [68].

### 4.6. Liver Disease

Currently, most of the literature regarding liver disease is related to liver cancer. However, there is an increasing interest in the application of HP ^13^C MRI in diagnosis and monitoring of compilations related to liver disease in pre-clinical studies [69,70,71,72]. Examples include the assessment of hepatic metabolism in non-alcoholic fatty liver disease (NAFLD) induced by a high-fat diet (HFD) in rats [73] or the effect of liver metabolism in inflammatory liver injury [74] and ethanol consumption as an early indicator of complications related to fatty liver disease, hepatitis, cirrhosis, and cancer [75]. Furthermore, assessment of metabolism in genetically modified knockout [76,77,78] and insulin-deficient rodent models [78,79] has been studied in detail. Like kidney, liver disease imaging is limited in human studies; however, one study has been reported on clinicaltrials.gov as “active, not recruiting” on the effect of fatty liver disease (Table A2). Recently a novel hyperpolarized probe, [2-13C]dihydroxyacetone (DHAc), has been applied to enable estimation of liver metabolism (gluconeogenesis, glycolysis, and the glycerol pathways), and this could very well be an important finding for further clinical attention [80].

### 4.7. Technical Advances

#### 4.7.1. Polarizer

Multiple methods to achieve hyperpolarization have been explored including brute force polarization, parahydrogen-induced polarization (PHIP)-based methods [81,82,83], and dynamic nuclear polarization (DNP). Currently, the only polarization process approved to be used for clinical studies of carbon-13 is DNP [11].

Brute force polarization is achieved by placing the probe in a strong magnetic field at a temperature close to 0 K [84]. While the method is straightforward to apply, it is very impractical and not useable for in vivo experiments given the temperature and very long T1 times in the solid state.

PHIP utilizes the change in parahydrogen and orthohydrogen spin states at low temperatures. Hydrogen molecules exists in two energy states—parahydrogen and orthohydrogen. At room temperature the spin states are 25% parahydrogen and 75% orthohydrogen; however, the spin states are almost complete parahydrogen (>99%) at very low temperatures (<20 K). Applying a catalyst, such as alkenes, breaks the symmetry of hydrogen, inducing a change in the spin order, known as PHIP [85]. Transfer of the polarization is thereafter performed by field cycling or polarization methods. PHIP has the advantage of being cheap and easy to use and allows for very rapid polarization. Several proposals to alter the PHIP process have been suggested, such as signal amplification by reversible exchange (SABRE [82]), synthesis amid the magnet bore allows dramatically enhanced nuclear alignment (SAMBADENA [83]), and PHIP by means of side arm hydrogenation (PHIP-SAH [86]). Until recently, the requirements of unsaturated precursor molecules, toxic solutions, and catalyzers have been a significant limitation for in vivo applications. However, PHIP-SAH may have found a way to overcome these limitations by using propargyl alcohol in the parahydrogen-induced polarization process.

DNP is based on the spin interaction of free electrons with nearby nuclear spins at low temperatures (<4 K). At this temperature the electrons have a low energy state and can achieve total polarization (100%). Applying irradiation with microwaves causes polarization exchange from the electrons to nearby nuclear spins. Electrons have a short T1 relaxation time causing them to return to their thermal equilibrium state and making them able to repeat the process on unpolarized nuclei. The process is repeated until a desired polarization is achieved according to the nuclei T1 relaxation time (polarization decreases in accordance with T1 relaxation time) [3].

Hyperpolarized carbon-13 using dissolution dynamic nuclear polarization was disseminated in 2003 by J. H. Ardenkjær-Larsen et al., demonstrating the so-called (“alpha” polarizer [3]). The technique was then commercialized in the form of HyperSense (Oxford Instruments, Abingdon, UK) and as the clinical 5 T SPINLab (GE Research Circle Technology Inc., GE Healthcare, Chicago, IL, USA) [87] in 2011. Furthermore, a new commercial 6.7 T pre-clinical polarizer has been introduced—the Spin-Aligner (Polarize IVS, Frederiksberg, DK) in 2018 [88]. The effectiveness of the polarizers has increased rapidly (from 20% to 40% to 70%), and clinical systems are able to achieve polarization up to 55% [89], which is twice the polarization used in the first clinical trial [7]. However, new technological and methodological advances are still needed to improve the polarization method, especially with respect to cost effectiveness and user friendliness [11,90,91].

#### 4.7.2. Sequences 

MR spectroscopic imaging is often preferred over single voxel spectroscopy to evaluate local changes in metabolism inside the organ of interest. The dimensionality of hyperpolarized MRSI data is higher, and one must consider possible contamination of signal from nearby voxels (signal bleed) [92], B0 field inhomogeneities across the organ, and increased data acquisition time. As the hyperpolarized signal is non-recoverable, rapid sequences are paramount to ensure collection of metabolic information. The hyperpolarized [1-13C]pyruvate MRSI experiment is composed of data in up to five dimensions (up to three spatial, one temporal, and one spectral) often leading to compromise in one or more dimensions to ensure rapid acquisition [93].

In proton MRI, a successful means of reducing acquisition time is to modify or sample fewer k-space points. These methods are, however, not directly applicable in MRSI. This is partly due to the inherent low number of sampling points, which do not introduce the required level of sparsity. Therefore, the focus of fast MRSI is on more efficient k-space sampling either as echo-planar imaging (EPI) [94] or spiral [95], radial [96], or concentric rings [97].

Fast imaging techniques can generally be grouped into three main approaches: (1) spectroscopic imaging; (2) prior knowledge model-based approaches; and (3) metabolite-specific imaging [98,99,100]. Acquisition time can be decreased with the combination of multiple receiver channel elements as in parallel imaging (SENSE [101], calibrationless [44]), compressed sensing [102,103,104,105], and multiband excitation [105,106] methods. Nevertheless, parallel imaging methods are challenging as acquisition of the needed sensitivity profiles is limited.

(1) Spectroscopic imaging includes phase-encoded chemical shift imaging (CSI). The CSI acquires a multivoxel MRS image, utilizing phase-encoding to achieve spatial resolution but at the cost of a longer acquisition time. 

(2) Prior knowledge model-based approaches are a faster imaging method often combined with spectroscopic imaging. The speed comes from efficient acquisition of k-space points and avoiding sampling of unnecessary signal, thereby utilizing prior knowledge of the substrates to improve temporal resolution or to reduce the required sampling matrix (SLIM [107], SLOOP [108], and SLAM [109,110]). There are several variations of the prior knowledge model-based techniques; however, recent developments with spectroscopic imaging by exploiting spatiospectral correlation (SPICE) and chemical shift encoding (CSE) could be preferred options for hyperpolarized carbon-13 MRSI.

SPICE utilizes a combination of a two acquisitions: first a set of low-spatial and high-temporal resolution data, followed by a set of high-spatial and low-temporal resolution data [111]. From the two acquisitions, high resolution spectroscopic images with an adequate spectral resolution are reconstructed [112]. The method has recently been evaluated in kidney models of mice [113]; nonetheless, results of the application of this novel technique will be interesting to follow in other organs.

CSE methods include Dixon or iterative decomposition with echo asymmetry and least squares estimation (IDEAL), which apply prior knowledge of the substrates and products [114]. CSE encodes the spectral dimension sparsely by acquiring only with a few different echo times (TEs) [115].

(3) Metabolite-specific imaging is the fastest imaging method of the three approaches, making it less sensitive to motion (shorter repetition time (TR)). However, the disadvantages are the requirement of a sparse spectrum and increased sensitivity to B0 field distortions [98]. The method rapidly acquires spectral and spatial data with frequency and slices selective RF pulses in an EPI [116] or spiral readout trajectory [5]. By exciting a single metabolite at a time and changing RF pulse resonance frequencies, dynamic datasets of desired metabolites are acquired.

#### 4.7.3. Data Acquisition, Reconstruction, Processing, and Analysis

Optimized data acquisition and processing steps can be vital in the following analysis. In this section, a pipeline for data acquisition, reconstruction, processing, and analysis is described. The pipeline may vary between institutions given the selection of coils, anatomy, and sequences (e.g., 2D EPI or 3D spirals). A schematic representation of the pipeline is illustrated in Figure 3, with indication of the conventional HP ^13^C MRI pipeline (light green) and suggestions for an extended pipeline (light grey).

##### Data Acquisition

B0 field map: Magnetic field inhomogeneity leads to a spread in resonance frequencies, causing difficulties preserving the spectral separation. The homogeneity requirements are determined by the metabolites of interest, and in HP ^13^C MRI, good separation is achieved when the B0 homogeneity is better than Δ0.1 ppm. Improvement of B0 field homogeneity is performed by shimming. Shimming complexity increases with the size of the field of view (FOV), and larger FOVs are more prone to magnetic field irregularities. Quality of the acquired free induction decay (FID) is the basis of signal analysis accuracy. Low SNR and frequency variations reduce the quality and result in poor spectra. Signal averages and larger volumes may increase the SNR, but this can be at the cost of resolution.

B1 transmit field map: Metabolite-specific B1 mapping of the transmit field of the carbon-13 tuned transmit coil should be performed. This provides a measure to correct for possible errors in the flip angles of the sequences used in data acquisition of fast HP ^13^C MRI sequences.

Fast MRS/MRSI sequence: A detailed description of HP ^13^C MRI sequences is given in a previous section.

##### Data Reconstruction

Coil combination: Combination of multiple receiver coils may introduce phase cancelation. This can be avoided by combining the signal acquired from the coils at each voxel either by weighted sum of squares, first point phasing, or singular value decomposition methods [43].

Phase and frequency correction can be manually performed in the data processing step to accommodate for signal loss due to phase cancellation with the use of zero- and first-order phase correction. Nevertheless, several spectral fitting approaches have included phase correction as part of the data processing pipeline (OXSA-AMARES [117], JMRUI [118], and LC model [119]), alleviating this as a data processing requirement.

k-space reconstruction: MRSI trajectories (Cartesian or non-Cartesian) are applied to reconstruct the acquired MRSI to ensure correct k-space gridding. A comprehensive review of HP ^13^C MRI sequences and reconstruction methods has recently been published by Gordon et al. [98].

##### Data Processing

Apodization filter: Apodization is used to enhance the SNR at the cost of spectral resolution (line broadening). The FID (time domain) or the spectra (frequency domain) is multiplied by a filter function, often an exponential function. 

Cramér–Rao lower bounds: Quality assurance of the spectra can be performed by evaluation of the metabolite separation, signal-to-noise ratio, and temporal resolution. One approach is to quantitatively evaluate the SNR, spectra line width, and spectra separation with the Cramér–Rao lower bounds (CRLBs) [120,121]. CRLBs can be used as a measure to determine voxels to be included or excluded before or even after metabolite spectral fitting.

Spectral line-shape correction: Spectral fitting algorithms consist of a combination of pre-determined line shapes (Lorentzian, Gaussian, or Voigt), though acquired signal line shapes may be distorted, e.g., due to magnetic field inhomogeneities and eddy currents. Line-shape correction can be performed by deconvoluting the spectra with a reference spectrum.

Spectral baseline correction: Acquired spectra may have shifts in the baseline and thereby overestimate the metabolite resonance peaks. The spectral baseline can be corrected by applying a low-order signal fit [122]. This leads to more robust data and improved data quantification.

Denoising: Denoising has attracted great interest in HP ^13^C MRI given the low SNR, good peak separation, and representation of metabolite peaks. Several applications of signal denoising have been evaluated for improvement of data quality, and recent applications of multidimensional tensor value decomposition show promising results by changing from a fixed rank [123] to an automatic cost function-based rank selection approach [43]. Nevertheless, denoising with the use of singular value decomposition (SVD)-based methods should be carried out with caution. The application could introduce oversimplification and thereby filter out lower SNR metabolites or disease metabolism as noise. Furthermore, if too few singular components (low rank) are used, kinetics of pyruvate could modulate the reconstructed dynamics/kinetics of the lower SNR metabolites.

##### Data Analysis

Data analysis is performed subsequent to data optimization in the acquisition and pre-processing steps. The multidimensional MRSI data consist of up to three spatial dimensions with a spectral and a temporal dimension (Figure 4A). Determination of spectral metabolites is performed by applying spectral fitting with the use of metabolite prior knowledge. The change in signal amplitudes in the temporal dimension is then used for evaluation of the metabolic flux as an estimate of the downstream metabolism from injected pyruvate to lactate, alanine, and bicarbonate (Figure 4B). Residual pyruvate may be detected in the spectrum as pyruvate hydrate. Analyses of the metabolic flux or exchange rates can be determined by fitting single-compartment or multicompartmental models. The forward rate constant of substrate conversion is calculated as an apparent conversion from pyruvate to, for example, lactate (kpl) [124,125] (Figure 4C). The results may, however, be prone to systemic errors if not combined with the pyruvate input function [126].

Alternatively, a model-free formulism based on the ratio of area under the curve (AUC) of the injected and downstream metabolite can be used. This has shown to be a robust and clinically relevant alternative to kinetic model-based rate measurements [127], especially in the lactate-to-pyruvate AUC ratio, which represents the full reaction as determined by compartment kinetic modelling [128]. In addition to being a simplified approach the benefit of evaluating the downstream signal of an AUC is to reduce bias in the later acquired time series [129]. In both approaches, kpl consistency should be evaluated. This could be done by only accepting kpl where the standard deviation is smaller than determined mean kpl. Furthermore, this approach may be beneficial in low SNR measurements for determination of pH from the ^13^CO_2_ to H^13^CO_3_^−^ ratio [130,131].

### 4.8. Other

It is crucial for the translation of basic science to clinical application that disease models are studied meticulously to ensure high precision and diagnostic value. This methodology is applied in pre-clinical studies with the use of cell models, hyperpolarized probe testing, and validation of widespread pathologies in animal models. Expanding the understanding of disease-altered metabolism and the underlaying processes involved in interventional treatment responses is key for method validation before initiating clinical trials.

This review focuses on the clinical transition; therefore, areas other than those listed in the previous sections (e.g., lung [132,133], angiography [134], placenta [135,136], muscle [137,138,139]), diseases (e.g., diabetes [63,140,141,142], rheumatoid arthritis [143,144], toxin-induced neuroinflammation [145], radiation injury [146,147]), and physiology (e.g., cell metabolism [148,149], pH [130], blood serum [150,151], bacteria metabolism [152]) are not be covered.

### 4.9. Clinical Transition

Hyperpolarized MRI is on the verge of clinical translation [1]; however; to ensure clinical adaptation, it is important to improve and validate the workflow. As of today, 50 polarizers prepared for injection into humans are installed worldwide [12], and more than 10 sites are performing clinical trials. More than 200 human subjects have participated in clinical trials with HP ^13^C MRI [1].

As of today, 17 clinical studies have been conducted (Table 1). The number of clinical studies is not a direct measure of the translational state; however, it does indicate the activities and research focuses of the hyperpolarized carbon-13 research community. Evaluating studies reported to clinicaltrials.gov as “active, not recruiting” or “recruiting” indicates that the field is pointing towards increase in clinical trials and covering a wider area of diseases. The number of clinical trials is set to double, with 27 new studies imminent (Table A2).

The database of interventional clinical trials with medical products in the European Union, EudraCT (European Union Drug Regulating Authorities Clinical Trials Database), lists an additional six HP ^13^C MRI studies, three of which were initiated during 2020 (Table A3).

The determination of the translational state is based on subjective measures rather than a systematic meta-analysis of the relatively low number of studies performed on a low number of subjects. Clinical transition occurs when a novel or improved diagnostic measure is needed. If conventional methods are superior, transition often does not occur. With this in mind, we focus on the translational state of five selected areas, highlighted in Table 1, and outline a qualified estimate of which will become a clinical application first.

Cardiac HP ^13^C MRI has been performed in clinical studies [5,54], while no kidney or liver studies have so far been published (excluding renal cancer). Three brain studies have been reported in the recent years on healthy brain [8,13,46]; however, none are reporting metabolic changes due to pathology. Therefore, we consider the current translational state of the four areas to be low to medium.

HP ^13^C MRI in cancer models has proven great potential, not only from the clinical trials but also from the possibilities of targeting cancer cell models in advanced experimental studies [1]. This provides a measure of going from specific cell culture analysis in small animal models to clinical trials with accurate and reproducible results. Transition from research to clinical implementation is plausible in prostate cancer given the large number of studies performed since the first clinical trial in 2013. Nevertheless, it has recently been proposed that there could be an even greater advantage of the application in breast cancer, potentially reducing the number treatments and treatment time (today approximately 12 treatments in 3 months) [4]. The level of detail and experience of HP ^13^C MRI in oncology is providing assurance of the value in the method; therefore, we consider the current translational state to be high.

### 4.10. Challenges and Limitations

HP ^13^C MRI is limited by the short lifetime of the hyperpolarized signal as well as the need to acquire five-dimensional data (three spatial, one temporal, and one spectral). The technology requires rapid imaging strategies to improve the quantification of metabolism. Research sites have improved dramatically in the effectiveness of producing consistent results and avoiding failed experiments, e.g., sample contamination, invalid or missing data. This is not reported in the literature but could be a useful guideline for the progress, complexity, and feasibility of HP ^13^C MRI as a future clinical method. Assessment of the literature shows most publications reporting technical advances; however, publications are focused on signal refinement over technological paradigm shifts. Therefore, the main limitation is believed to be in the clinical workflow and not the MR methods.

Several challenges can arise regarding the dissolution workflow of HP ^13^C MRI, ranging from the costly components, substrates, and laboratory quality assurance. This combined with long dissolution travel time results in unnecessary loss of signal, as the transverse relaxation time is only a matter of minutes, complicating the chances of successful examinations. Nevertheless, recent advances show the ability to reduce the hyperpolarized substrate travel time from the polarizer to injection, and a few sites have achieved results of 30 s or less [4]. Another approach is to use UV-generated labile free radicals [154]. This method can be used to create nuclear polarization storage and transport the samples across larger distances before being applied. Hyperpolarized carbon-13 pyruvate is commonly produced via the dDPN; however, recent discoveries indicate the possibility of producing hyperpolarized substrates via PHIP-SAH [86]. This could be a cheaper and faster alternative to produce hyperpolarized substrates and thereby accelerate the transition to clinical application.

Improved workflow may come from consensus and multicenter studies. Multicenter studies [155] and detailed descriptions of the methods used in data acquisition and analysis [156] of clinical HP ^13^C MRI experiments are needed to ensure reliability and comparability across research sites. Although several sites are performing clinical trials, the number of study participants is still limited. Improved comparability should strengthen the evaluation of clinical studies and alleviate possible bias occurring from study design differences. Nonetheless, a recent study focused on the development of methods and feasibility of using HP ^13^C MRI data for evaluating brain metabolism to guide the community towards comparability [157]. If reports such as this are conducted in other areas, it could prove beneficial to creating consensus and furthermore helping initiatives of new research sites.

Since the spin-lattice relaxation time (T1) is affected by the system field strength, it could be proposed to increase T1 by lowering the field strength. However, this approach would reduce the ability to separate metabolites by chemical shift by a factor of two, when shifting from a 1.5 to 3 T MRI system. Therefore, it is expected that transition to clinical practice will be on 3 T systems, a statement supported by the current application in clinical studies (clinical trials (Table A2) and EudraCT (Table A3)).

## 5. Conclusions

The transition of hyperpolarized carbon-13 to clinical applications has been debated regarding clinical needs and technological advancements. This review evaluates the current state of HP ^13^C MRI through a comprehensive literature analysis with emphasis on the road to clinical application. This review highlights the movement of the community towards multicenter trials, with an immense increase in the number of clinical trials being performed in the coming years. The conclusion of this review is that the workflow of HP ^13^C MRI is the limiting factor to achieving clinical application.

## Figures and Tables

**Figure 1 metabolites-11-00219-f001:**
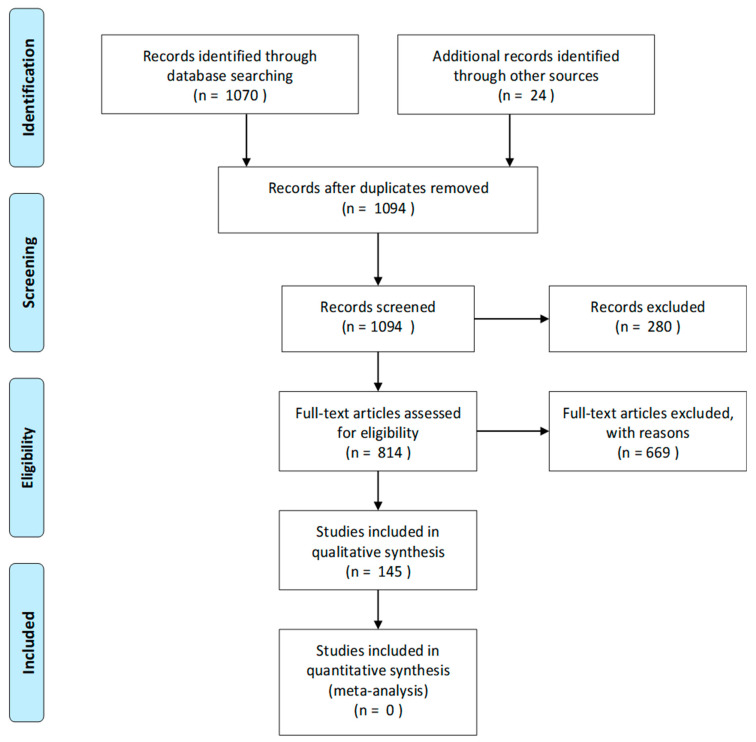
PRISMA 2009 flow diagram of the paper selection process.

**Figure 2 metabolites-11-00219-f002:**
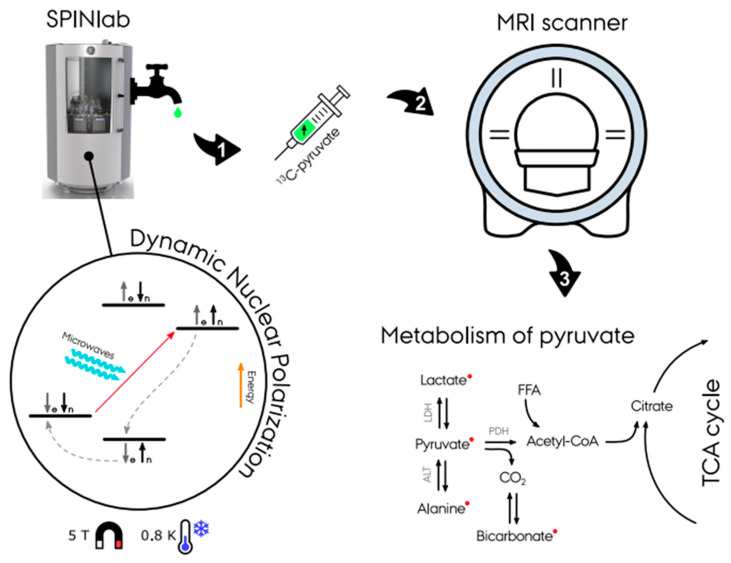
Illustration of the hyperpolarizing experiment in a SPINlab dissolution dynamic nuclear polarizer (d-DNP). First, the dissolution of ^13^C-pyruvate is placed in a 0.8 K cold environment at 5 T within the DPN to achieve hyperpolarization. Secondly, the dissolution achieves massively increased magnetic properties and is ready for injection into the experiment subject. With the use of fast MR sequences, spectra and images of the pyruvate metabolism is achieved (Courtesy of Christian Ø. Mariager).

**Figure 3 metabolites-11-00219-f003:**
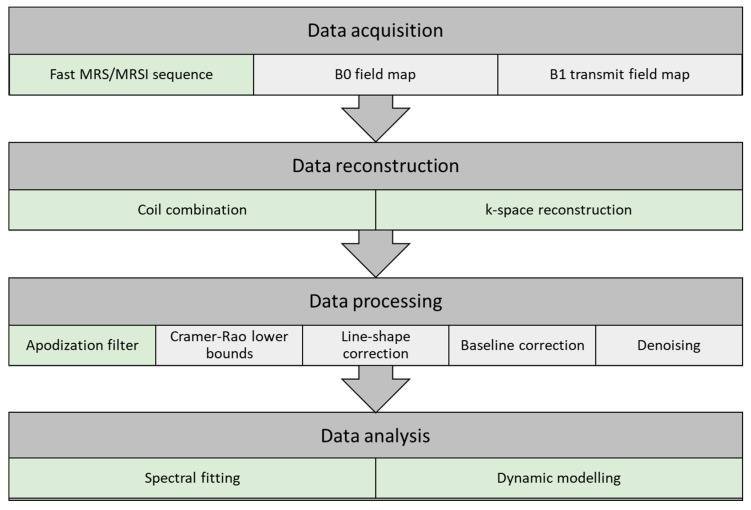
Flow chart of a pipeline for data acquisition, reconstruction, processing, and analysis of hyperpolarized carbon-13 MR spectra. The pipeline illustrates the conventional approach (light green) and suggestions for extended steps (light grey).

**Figure 4 metabolites-11-00219-f004:**
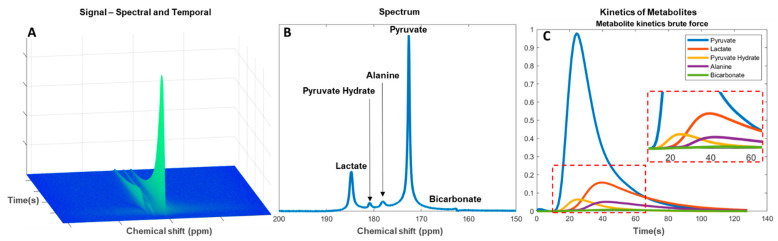
Analysis of hyperpolarized carbon-13 spectra from pyruvate injection. (**A**) Metabolite signal change in the spectral and temporal dimension since injection. (**B**) Total signal of downstream metabolites in a temporal summed spectrum. (**C**) Kinetics of the metabolites with enhanced view of smaller signal metabolites (red dashed box).

**Table 1 metabolites-11-00219-t001:** Overview of applications for which clinical trials have been performed with hyperpolarized carbon-13 MRI and their translation state towards clinical adaption.

Area of Interest	Publications	Human Trials	Translation State
Oncology	170	prostate: [7,20,22,23,24] pancreas: [27], breast: [28,29], brain: [9,20,32,153], kidney: [33]	High
Brain	37	[8,13,46]	Low
Heart	87	[5,54]	Medium
Kidney	31	-	Low
Liver	21	-	Low
Technical advances ^1^	349	-	-
Other ^2^	118	-	-

^1^ Sequence, polarization, coils, etc. ^2^ Other anatomical studies (e.g., lung, angiography, placenta, muscle), diseases (e.g., diabetes, rheumatoid arthritis, toxin-induced neuroinflammation, radiation injury), and physiology (e.g., cell metabolism, pH, blood serum, bacteria metabolism).

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
