# Peer review of "Comprehensive Literature Review of Hyperpolarized Carbon-13 MRI: The Road to Clinical Application"

_metabolites, 2021, doi:10.3390/metabo11040219_

Round 1

Reviewer 1 Report

Line 114-115 - the sentence herein requires some restructuring to make sense such as “…conversion of pyruvate into lactate, a metabolic shift which is often very different from…”

Line 129 - “animals is…” should be “animals are…”

Line 131 - “preferer” should be “prefer”

Line 134 - “is” should be “are”. — subject is plural in this sentence

Line 147-148 - the meaning of the sentence “Achievement of success is made…” is unclear; authors should clarify the verbiage here.

Line 152 - the word “attentive” should perhaps be replaced by “intention” ?

Line 160 - “pyruvate have been demonstrated” should be  “pyruvate studies have been demonstrated”

Line 163 - “conflicting” should probably be “contra-indicated”

Line 170 - “and additionally has shown potentials in” should be “and, additionally, have shown potential with”  

Line 171-172 - “Alternative,” should be “Alternative” (no comma) and “holds” should be “hold”

Line 193 - “only polarizer approved” should be “only polarization process approved”

Line 207 - “has” should be “have”

Line 219 - “a unpolarized” should be “an unpolarized”

Line 243 - “fever” should be “fewer”

Line 346 - “an fixed rank” should be “a fixed rank”

Line 348 - “used with caution, the application could” should be broken into two sentences — “used with caution. The application could”

Line 350 - “are used kinetics of pyruvate” requires a comma as it is an ‘if’ phrase — “are used, kinetics of pyruvate”

Line 353 - “are” should be “is”; the subject ‘analysis’ is singular

Line 354 - “op” should be “up”

Sentence in Line 450-451 — this statement seems somewhat contradictory to statement immediately preceding it; think it needs to be revisited and perhaps reworded with at focus on what is meant by “clinical workflow”.  Workflow is further elaborated in subsequent paragraphs, but this is not truly “clinical” workflow. Perhaps “dissolution workflow” would be a better term.

Line 472 —“If reports like this is conducted”; “is” should be “are”

Round 2

Reviewer 2 Report

Well done